# Three-Dimensional Porous Scaffolds Derived from Bovine Cancellous Bone Matrix Promote Osteoinduction, Osteoconduction, and Osteogenesis

**DOI:** 10.3390/polym13244390

**Published:** 2021-12-15

**Authors:** Alda Malagón-Escandón, Mathieu Hautefeuille, Edgar Jimenez-Díaz, Jesus Arenas-Alatorre, José Manuel Saniger, Isidro Badillo-Ramírez, Nadia Vazquez, Gabriela Piñón-Zarate, Andrés Castell-Rodríguez

**Affiliations:** 1Facultad de Medicina, UNAM, Mexico City C.P. 04510, Mexico; aldamalagon@icloud.com (A.M.-E.); nadisva@ciencias.unam.mx (N.V.); gabrielapinon@unam.mx (G.P.-Z.); 2Facultad de Ciencias, UNAM, Mexico City C.P. 04510, Mexico; mathieu_h@ciencias.unam.mx (M.H.); edgarjd@ciencias.unam.mx (E.J.-D.); 3Instituto de Física, UNAM, Mexico City C.P. 04510, Mexico; jarenas@fisica.unam.mx; 4Instituto de Ciencias Aplicadas y Tecnología (ICAT), UNAM, Mexico City C.P. 04510, Mexico; jose.saniger@icat.unam.mx (J.M.S.); ibadillo@ciencias.unam.mx (I.B.-R.)

**Keywords:** three-dimensional porous scaffolds, bone extracellular matrix, bone regeneration, tissue engineering, biomaterials

## Abstract

The use of three-dimensional porous scaffolds derived from decellularized extracellular matrix (ECM) is increasing for functional repair and regeneration of injured bone tissue. Because these scaffolds retain their native structures and bioactive molecules, in addition to showing low immunogenicity and good biodegradability, they can promote tissue repair and regeneration. Nonetheless, imitating these features in synthetic materials represents a challenging task. Furthermore, due to the complexity of bone tissue, different processes are necessary to maintain these characteristics. We present a novel approach using decellularized ECM material derived from bovine cancellous bone by demineralization, decellularization, and hydrolysis of collagen to obtain a three-dimensional porous scaffold. This study demonstrates that the three-dimensional porous scaffold obtained from bovine bone retained its osteoconductive and osteoinductive properties and presented osteogenic potential when seeded with human Wharton’s jelly mesenchymal stromal cells (hWJ-MSCs). Based on its characteristics, the scaffold described in this work potentially represents a therapeutic strategy for bone repair.

## 1. Introduction

The bone extracellular matrix is formed by a structural support network of type I collagen, noncollagenous proteins, lipids, and growth factors [1].

When the bone is injured, extracellular matrix components (ECM) allow cell adhesion, migration, proliferation, and differentiation at injury sites. Additionally, the ECM structure interconnected pores favor adequate vascularization that confers the self-healing ability to bone [2]. Moreover, the ECM regulates cell–matrix and cell–cell interactions and angiogenesis based on signaling induced by growth factors and cytokines [2,3].

However, critically sized bone defects cannot heal on their own and represent a challenge due to their dimensions, geometric complexity, damage to surrounding soft tissues, and other patient conditions, such as age and general health [4]. The gold-standard treatment of critical-sized bone defects is the use of autologous bone grafts [5]. Autologous bone grafts, mainly composed of extracellular matrix and cells, must be obtained in sufficient quantity and be capable of promoting the healing of the lesion. Nonetheless, these grafts present several disadvantages since they are contraindicated in patients with systemic diseases or comorbidities, also related to the complications and costs of intervening in two surgical areas [5]. As an alternative, allogeneic and xenogeneic bone grafts (mainly bovine) have been approved by the Food and Drug Administration (FDA) for use in humans [6]. Still, their clinical success rate is unpredictable [7]; this is mainly due to mineralized grafts; they hinder the balance between bio absorption and volume maintenance to achieve ideal bone regeneration [8,9,10,11,12,13,14], and the immune response associated with cells, cellular debris, and collagen [7,12]. For this reason, xenografts should undergo different processes that allow maintaining their structure, ECM composition, function, and bioavailability of growth factors because all these characteristics are critical in bone regeneration. For all the above, bovine bone xenografts processing should include the following:(1)Decellularization to eliminate cells and immunogenic determinants.(2)Demineralization to eliminate the mineral phase; and(3)Hydrolysis of collagen to decrease the likelihood of foreign body granuloma and immunogenic response and favor the degradation rate.

Successful decellularization of organs and tissues aims to eliminate cells, remove genetic material, preserve components such as growth factors, glycosaminoglycans, noncollagenous ECM proteins, morphology, and mechanical properties [8].

Demineralization eliminates mineral ions from the bones, and it is generally carried out with hydrochloric acid between 0.5 and 0.6 M [9,10,11,12,13,14,15]. It should be clarified that these concentrations of HCl can modify the morphological, mechanical, and biological properties of the ECM; however, no studies compare characteristics of scaffolds demineralized with low concentrations of HCl, so it is necessary to evaluate them before considering clinical translation [9].

On the other hand, there is no consensus about a possible immune response by type I collagen. Various studies have described the properties of native type I collagen and hydrolyzed type I collagen [16,17,18,19,20]. The hydrolysis of collagen is an irreversible conformational change of the protein into compounds with relatively lower molecular weights depending on the time and concentration of the denaturing agent. Hydrolyzed collagen exhibits increased hydrophilicity, low immunogenicity, a nonallergenic response, faster degradation, and improved bioactivity because it exposes cryptic regions, such as the RGD (Arg-Gly-Asp) motifs recognized by CD51 expressed in MSCs derived cells such as osteoblasts [21,22,23,24].

The efficacy of bovine bone xenografts has been questioned due to the mineralization, presence of cells, immunogenic determinants, deproteinization, loss of morphology, and mechanical properties of the ECM. However, these disadvantages could be solved with different strategies of decellularization, demineralization, and hydrolysis methods that allow preserving the morphology, composition, and mechanical properties of the ECM.

This study addresses obtaining scaffolds from bovine cancellous bone by decellularization, demineralization, and collagen hydrolysis, using two HCl concentrations lower than those used in other studies and alkaline hydrolysis. The scaffolds obtained were characterized biologically, chemically, histologically, and mechanically. Furthermore, the scaffolds’ osteoinductive, osteoconductive, and osteogenic potential was tested in vitro by culturing hWJ-MSC.

## 2. Materials and Methods

### 2.1. Obtaining of the Scaffolds

Cancellous bone blocks of 1 cm^3^ were obtained from the epiphysis of bovine femurs (18 to 24 months of age). The bones were obtained from a local slaughterhouse. Cancellous bone blocks were stored at −80 °C until use. Blood and lipids were removed before processing. To remove the blood, the blocks were washed thoroughly using distilled water, immediately submerged in 5% NaOCl solution (Grupo Dimex, Mexico City, Mexico), incubated for 24 h under constant stirring at 100 rpm at 37 °C, and then washed thrice with PBS at 4 °C.

To remove lipids from the bone blocks, the blocks were immersed in 100 mL of a 1:1 solution of methanol-chloroform (Sigma Aldrich Inc., Saint Louis, MO, USA) for 24 h at room temperature and then immediately washed with distilled water thrice. Afterward, distilled water was added at 60 °C, and the blocks were allowed to return to room temperature for 1 h. The bottle was kept open in a fume hood during this procedure.

Due to the complexity of the composition of bone tissue, it is necessary to perform decellularization and demineralization of the bovine bone blocks and hydrolysis of collagen protein. The next protocol was applied:

(1) Ten grams of bovine bone block was placed in 100 mL of dodecyl sulfate sodium solution (SDS) and 1% Triton X-100 solution (Sigma Aldrich Inc., Saint Louis, MO, USA) for 24 h at 37 °C under constant stirring in a shaking water bath (model SWBR17, MRC lab Ltd., Ha-Gavish, Holon, Israel). At the end of the process, the blocks were washed thrice with PBS at 4 °C.

(2) The blocks were demineralized using 37% hydrochloric acid solution (Sigma Aldrich Inc., Saint Louis, MO, USA) at three different concentrations (0.5 M, 0.250 M, and 0.125 M). Then 10 grams of bovine bone blocks were placed in 100 mL of every HCl solutions. Constant stirring at 100 rpm was performed for 24 h at 37 °C in a shaking water bath (model SWBR17, MRC lab Ltd., Ha-Gavish, Holon, Israel). The bovine bone blocks were washed with distilled water and PBS at 4 °C to remove excess acid. Immediately, the bone blocks were immersed in PBS and refrigerated for 2 h at 4 °C and then washed thrice with PBS at 4 °C.

(3) Finally, alkaline hydrolysis of the extracellular matrix of decellularized and demineralized bovine bone was performed. One hundred milliliters of 1 N NaOH solution (Sigma Aldrich Inc., Saint Louis, MO, USA) was used for every 10 g of bovine bone block. Constant agitation was performed at 100 rpm for 24 h at room temperature in a shaking water bath (model SWBR17, MRC lab Ltd., Ha-Gavish, Holon, Israel) and washed thrice with PBS at 4 °C. The extracellular matrix of decellularized and demineralized bovine bone and the hydrolyzed collagen protein processed in three different concentrations of HCl were termed 0.5 M scaffold, 0.25 M scaffold, and 0.125 M scaffold according to the HCl concentration because these concentrations are the only variables in the method of obtaining scaffolds.

The scaffolds were vacuum freeze-dried for 24 h, immediately sterilized with a UV ozone cleaner (UV Ozone Cleaner ProCleaner™ Plus model, BioForce Nanosciences, Inc., Virginia Beach, VA, USA) for 30 min, and stored in airtight packages until use.

### 2.2. Characterization of the Scaffolds

To verify complete cell removal, the scaffolds were dyed with 4′,6-diamidino-2-phenylindole (DAPI) and Masson’s trichrome staining. The scaffolds and bovine bone blocks were fixed in 4% paraformaldehyde solution (Sigma Aldrich Inc., Saint Louis, MO, USA) overnight, paraffin-embedded and cut into 10-μm thick sections. Finally, sections were stained with Masson’s trichrome (Sigma Aldrich Inc., Saint Louis, MO, USA). Additionally, the scaffolds as well as the bovine bone without processing and the commercial scaffold Cancellous Bone Chips Biograft^®^ were stained with DAPI (1:1200; Sigma Aldrich Inc., Saint Louis, MO, USA). Then, the histologically stained sections were imaged at 40X magnification with an Olympus I optical microscope (Model X50, Olympus Inc., Tokyo, Japan) and a Leica confocal microscope (model TCS SP8, Leica Inc., Wetzlar, Germany). Then, using Image-Pro, we analyzed the photomicrographs stained with Masson’s trichrome and quantified the mineralized areas in each scaffold and bovine bone (*n* = 3).

To analyze the three-dimensional ultrastructural features of the scaffolds, scanning electron microscopy (SEM) coupled with a dispersive energy system (Model 5600LV, Jeol Ltd., Akishima, Tokyo, Japan) under a low vacuum with the backscattered electron technique was used at an acceleration of 15 kV voltage and a magnification of 50× according to the scale bar, the average pore size of the bovine bone and the scaffolds was estimated from the SEM digital images obtained by ImageJ software.

SEM/EDS analysis was performed using SEM coupled with a dispersive energy system (Model 5600LV, Jeol Ltd., Akishima, Tokyo, Japan) at an accelerating voltage of 20 kV and a working distance of 10 mm. Scaffolds and bovine bone blocks were placed under low vacuum and subjected to EDS point analysis with acquisition times of 10–30 s.

The scaffolds were characterized by Raman spectroscopy using a WITec Alpha 300 R Raman-AFM confocal microscope (WITec Inc., GmbH, Ulm, Germany). Measurements were performed using 532 nm laser light excitation, from a Nd: YVO4 incident laser beam, with a power of 3.56 mW and detection of 672 lines/mm grating. The 1-cm^3^ scaffolds were sectioned longitudinally with a scalpel blade for the analysis of the internal regions, and samples were mounted on cover glass for analysis. Spectra were collected through a 20× objective with a 3-s integration time and 50 accumulations. In addition, Raman spectra were collected of type I hydrolyzed collagen in powder without any treatment and measured under the same conditions.

As a complement to Raman spectroscopy, Fourier transform infrared (FT-IR) characterization was performed using a spectrometer equipped with an attenuated total reflectance (ATR) diamond crystal (model Nicolet iS50R, Thermo Scientific, Inc., de Waltham, MA, USA). Spectra were collected with 32 scans and with a 4 cm^−1^ spectral resolution in the range of 400–4000 cm^−1^. Measurements were performed in triplicate per group.

The elastic behavior of the 1-cm^3^ scaffolds was determined using the microindentation technique with an FT-MTA-03 microindenter system equipped with an FT-S200 sensor and a 50-µm diameter spherical tip (force range ±200 µN) under hydrated conditions in 1 mL of Dulbecco’s Modified Eagle Medium: Nutrient Mixture F-12 (DMEM-F12) culture medium for 5 min. Then, nine microindentation tests were performed on each sample at room temperature.

The results were used to calculate the Young’s modulus of the scaffolds. As a control, commercial scaffolds Cancellous Bone Chips Biograft^®^ were used. We set the Poisson ratio to 0.5 to calculate the Young’s modulus of the scaffolds using the following equation:$$E=\frac{3}{4\sqrt{R}}\;\nu \delta^{-1.5}$$
where *E* is the Young’s modulus, *R* is the radius of the tip, *δ* is the indentation depth and *ν* is the Poisson ratio.

The swelling ratio of the scaffolds was determined using the standard gravimetric method. First, the dry weight (*Wd*) was determined. Then, to determine the wet weight (*Ws*), the material was placed in a conical tube, and 5 mL of PBS was added at pH 7.4. Finally, the sample was incubated for 5, 24, 48, 72, 96, and 168 h at 37 °C. The wet scaffolds were removed from the conical tubes and placed for 10 s on filter paper to remove the excess PBS; next, we weighed each scaffold.

The swelling index of the 1-cm^3^ scaffolds was calculated based on the ratio of the weight increase to the initial weight. Each value in triplicate was averaged, and then the following equation was employed to determine the swelling ratio:$$Swelling\;ratio\;\left(\%\right)=\frac{\left(Ws-Wd\right)\;}{Wd}\times 100$$

The degradation time of the 1-cm^3^ scaffolds was determined using the standard gravimetric method. One hundred milliliters of ethane sulfonic acid (HEPES) (Sigma Aldrich Inc., Saint Louis, MO, USA) were prepared, and then 100 μg of type I collagenase and 0.1% lysozyme solution were added (Sigma Aldrich Inc., Saint Louis, MO, USA.). First, the initial weight (*Wi*) was determined, and the scaffolds were immediately placed in conical tubes and immersed in the degradation buffer described above. The scaffolds were kept at 37 °C for 24 h and continuously stirred in a shaking water bath (model SWBR17, MRC lab Ltd., Ha-Gavish, Holon, Israel). Later, the scaffolds were weighed every 24 h to determine the final weight (*Wf*).

The percentage of weight loss was used as an indicator of degradation. Additionally, three measurements per group were examined every day for 21 days based on the following equation:$$Weight\;loss\;\left(\%\right)=\frac{Wi-Wf}{Wf}\times 100$$

### 2.3. hWJ-MSCs Isolation from Umbilical Cords

The donors signed informed consent previously, and we carried out the collection according to the guidelines of the ethics and research committees of the Faculty of Medicine of the National Autonomous University of Mexico (29-2016). Three umbilical cords from pregnancies obtained by cesarean section were placed in sterile bottles with sufficient culture medium and then transferred to the laboratory for refrigeration. Next, the transport medium was removed, and the umbilical cords were washed with sterile PBS thrice to remove the blood cells and immediately washed with a sterile Hanks solution. Finally, the umbilical cords were cut into small pieces of 1–2 mm^3^ using a sterile scalpel. The fragments were placed in a Petri dish for 2 weeks in 1:1 DMEM-F12 (BioWest Inc., Rosenberg, TX, USA) supplemented with 10% fetal bovine serum (FBS; BioWest Inc., Rosenberg, TX, USA) and 100 mM penicillin-streptomycin (BioWest Inc., Rosenberg, TX, USA) for 2 weeks. Then, the explants were removed from the dish.

When the cell culture was established, viable cells were counted based on the trypan blue test. The cells were seeded in 75-cm^2^ cell culture flasks (Falcon^®^, Coring, Inc., New York, NY, USA). When the culture reached 80% confluence, the cells were detached with a trypsin/ethylenediaminetetraacetic acid (EDTA) solution 0.05% (Gibco^TM^ Thermo Fisher Inc., Waltham, MA, USA). Afterwards, cytocompatibility was evaluated, and the concentration was adjusted. The cells were transferred to 75-cm^2^ cell culture flasks (Falcon^®^, Coring, Inc., New York, NY, USA) at a density of 2 × 10^3^/cm^2^. The hWJ-MSCs were expanded until reaching 80% confluence at passage 3.

### 2.4. hWJ-MSCs Characterization

The hWJ-MSCs that were used were obtained from passage 3 at 80% confluence. The cells were detached from cell culture flasks (Falcon^®^, Coring, Inc., New York, NY, USA) using trypsin/ethylenediaminetetraacetic acid (EDTA) solution 0.05% (Gibco^TM^ Thermo Fisher Inc., Waltham, MA, USA) and washed with PBS. Then, the hWJ-MSCs were incubated for 30 min at room temperature with saturating concentrations of the following monoclonal antibodies labeled with fluorochromes: PerCP/Cy5.5 anti-human CD 105, FITC anti-human CD90, APC anti-human CD 73, PE anti-human CD 34, PE anti-human CD 45, PE anti-human CD19, PE anti-human CD 11b, PE/Cy7 anti-human HLA-ABC, and PE anti-human HLA-DR (all purchased from BD Biosciences (San Jose, CA, USA). After removing the excess antibody through 3 washes with PBS, the hWJ-MSCs were ana-lyzed according to standard protocols using a flow cytometer (FACS Calibur™, BD Biosciences, San Jose, CA, USA) and FlowJo software (Becton, Dickinson & Company, Franklin, NJ, USA). The cell population was identified, and whether cells were positive or negative for each marker was determined based on the criteria of the International Society for Cell and Genetic Therapy (ISCT).

### 2.5. hWJ-MSCs Culture

For experimental studies, previously isolated and characterized hWJ-MSCs were used at passage 3. Cells were maintained in 1:1 DMEM-F12 (BioWest Inc., Rosenberg, TX, USA) culture medium supplemented with 10% fetal bovine serum (FBS; BioWest Inc., Rosenberg, TX, USA) and 100 mM penicillin-streptomycin (BioWest Inc., Rosenberg, TX, USA). The culture medium was changed every 2–3 days. The hWJ-MSCs were incubated at 37 °C with an atmosphere of 95% oxygen and 5% CO_2_.

### 2.6. In Vitro Assays

Cell attachment to the scaffolds was determined by immunohistochemistry with the polyclonal antibody anti-CD51/ITGAV (Enzo Life Sciences, Inc., Farmingdale, NY, USA).

For cell culture, the scaffolds were placed inside a Biosafety Cabinet Class II (model IIA2-X, Ecoshel, Notre-Dame-de-Stanbridge, Québec, Canada) with sterile tweezers. Immediately, cells were transferred to a 96-well plate and seeded with 10,000 hWJ-MSCs on the scaffolds. The cells were maintained for 7 days in culture with DMEM-F12 medium. Subsequently, the scaffolds were washed with PBS, fixed in 4% paraformaldehyde solution overnight, paraffin-embedded, and cut into 10-μm thick sections.

For the immunohistochemical analysis, histological sections of the scaffolds were deparaffinized and dehydrated. Bovine bone blocks without processing were used as controls. First, the sections were washed with citrate buffer at pH 3–3.5. Next, antigenic recovery was performed in a pressure cooker at 120 °C for 3 min. Immediately, endogenous peroxidase was blocked with 3% H_2_O_2_. Samples were incubated overnight with the primary antibody. The sections were washed three times with PBS, and the secondary antibody was applied. Finally, 3,3′-diaminobenzidine (DAB) was added, and samples were counterstained with H&E.

The stained sections were imaged with an Olympus I microscope (Model X50, Olympus Inc., Tokyo, Japan) at 40× magnification. Microphotographs were obtained in triplicate for each group. Then, using Image-Pro, we analyzed the photomicrographs to quantify the expression of CD51 in each scaffold and bovine bone (*n* = 3).

The cytocompatibility of the scaffolds was evaluated by seeding them with 10,000 hWJ-MSCs. The scaffolds with the cells were incubated for 1, 3, 7, 14, and 28 days at 37 °C and 5% CO_2_ with a change in culture medium every 2–3 days.

At the end of each experiment, each of the wells was washed with culture medium. After completing each experiment, the live/dead kit was used, and cells were incubated in an atmosphere of 95% oxygen and 5% CO_2_ for 30 min at 37 °C. In addition, 10,000 hWJ-MSCs were seeded on two coverslips and a live/dead test was performed. In addition, 70% ethanol was intentionally added to the coverslip to kill the hWJ-MSCs. The scaffolds were counterstained with DAPI to visualize the cell nuclei, which facilitated identification of the characteristics of living and dead hWJ-MSCs in scaffolds.

Before microscopic evaluation, the scaffolds were removed from the culture flask and then placed on slides with a drop of PBS. The slides were then viewed with a confocal microscope.

Images were obtained in triplicate for each group at different time points with a Leica confocal microscope (model TCS SP8, Leica Inc., Wetzlar, Germany). The different scaffolds with hWJ-MSCs were examined, and we counted the number of alive or dead cells (ImageJ-Cell Counter). Representative images of each condition were chosen from the digital images obtained by ImageJ (National Institutes of Health, Bethesda, MD, USA).

The capacity of the scaffolds to induce mineralization was evaluated by seeding 10,000 hWJ-MSCs and culturing them for 14 and 28 days in culture with DMEM-F12 medium with medium change every third day. In these cultures, there is no used mineralization-inducing medium. Subsequently, the scaffolds were washed with PBS and fixed with 4% paraformaldehyde, and 10-µm histological sections were obtained that were stained using the von Kossa method. In addition, photomicrographs at 40× were obtained in triplicate for each group. The images were analyzed, and we quantitatively (with the Image-Pro program) determined the degree of mineralization of the scaffold based on the color changes associated with the calcium phosphate deposits.

### 2.7. Statistical Analysis

The three different groups of scaffolds were analyzed in triplicate in each experiment except for the largest pore diameter variable, which was examined based on 50 measurements per group. Raman and FT-IR spectroscopy were used to examine nine scaffolds per group, and mechanical tests were used to obtain 27 measurements per group.

The dependent variables included the largest pore diameter, Young’s modulus, swelling ratio, degradation time, and HCl concentration at which the scaffolds were obtained. The independent variable was the experimental evaluation time interval.

The absolute values obtained for the statistical analysis and the experimental data are reported as the mean ± standard deviation. The GraphPad Prisma 7 program was used to perform a two-way ANOVA to identify the differences between the control group (bovine bone without processing or commercial scaffolds Cancellous Bone Chips Biograft^®^) and three experimental groups (0.125 M, 0.25 M, and 0.5 scaffolds). Finally, the post hoc Tukey test was performed for multiple comparisons, and statistical significance was considered at *p* < 0.05.

## 3. Results and Discussion

### 3.1. Characterization of the Scaffolds

Microscopic analysis of the scaffolds stained with DAPI, and Masson’s trichrome staining revealed that the scaffolds did not present cell nuclei or nuclear material; however, the bovine bone and commercial scaffolds Cancellous Bone Chips Biograft^®^ presented cell nuclei or nuclear material and intense blue marks that can be attributed to the detection of dsDNA (Figure 1).

Due to the complexity of the bone tissue [1], to obtain the scaffolds, we performed freeze-thaw cycles and mechanical agitation. Then, tissues were exposed to hydrochloric acid, Triton X-100, sodium dodecyl sulphate, ethylenediaminetetraacetic acid, and sodium hydroxide. Moreover, low concentrations were used for the ionic and non-ionic surfactants SDS and Triton X-100, respectively [25,26].

The effect of each technique and reagent on the decellularization determined through DAPI staining provided sufficient evidence; thus, whether the scaffolds had less than 50 ng of dsDNA per mg of dry weight [10], did not have to be determined because the 0.5 M, 0.25 M, and 0.125 M scaffolds did not exhibit any evidence of nuclear material or dsDNA. Our evidence demonstrates that the decellularization methods used here are effective for all scaffolds [10].

Masson’s trichrome staining revealed the organic and mineralized matrix in scaffolds. The blue staining on the scaffolds 0.125 M and 0.25 M suggests that the collagen fibers were well preserved, but in the scaffolds 0.5 M, the collagen fibers were parallel but separate (light blue). The color difference was related to collagen with a higher degree of denaturation. The red sections depict matrix mineralization. Minimal red zones are noted in the scaffolds 0.125 M 22.11% ± 1.3, 0.25 M 17.55% ± 4.7, and 0.5 M 4.95% ± 0.12, indicating low mineralization * *p* < 0.0001 compared with the bovine bone mineralized matrix. As we expected, the bovine bone not subject to processing was mineralized 81.71% 5.12 (Figure 2). The experimental findings were as expected [8,16,27], and the more obvious structural changes can be attributed to a higher concentration of HCl [28,29]. These results indicate that the structure of the ECM was preserved after performing the methods to obtain the scaffolds.

The scaffolds retained the three-dimensional structure, macroporosity, and interconnected macropores based on the SEM analysis. The trabeculae increased in width and the pore diameter was reduced as the HCl concentration increased (Figure 3). The diameters of the pores in all scaffolds were greater than 300 μm. The pore diameters were 356.69 μm ± 125.36, 352.82 μm ± 96.65, and 300.08 μm ± 94.96 in the 0.125 M, 0.25 M, and 0.5 M scaffolds, respectively. We used bovine bone not subject to processing as a control, and the pores were 372 μm ± 141.61 (Figure 3). The three-dimensional structure is one of the most important characteristics of cancellous bone, and it is essential for adequate cellular communication, vascularization, nutrient exchange, and osseointegration of the scaffold. The porosity of the materials for tissue engineering applications is critical, particularly in the skin and musculoskeletal, because it influences cellular activities, such as cellular adhesion, proliferation, vascularization, and diffusion of nutrients and metabolites [30]. Macropores are essential for the clinical success of scaffolds because a pore size of ~300 to 500 μm has been shown to allow the growth of neoformed bone and blood vessels. On the other hand, decellularization and decalcification methods consistently maintained pore size. The foregoing information is relevant to ensure adequate intercellular communication and extracellular fluid perfusion [31,32].

Several authors have proposed that the pore diameter of a bone scaffold should ideally be between 300 and 1300 μm and that 90% of the pores should be interconnected throughout the structure [33,34]. Nevertheless, the porosity range and the interconnection of pores are difficult characteristics to recreate in manufactured materials [30,31], due to the difficulty of reproducing three-dimensional structures [32,35], and the shrinkage of materials in certain cases. In contrast, the method used to obtain hydrolyzed collagen scaffolds from cancellous bovine bone was highly reproducible. The pores maintained a constant size, and the structure was not lost, which contrasts with that noted with other methods.

A demineralized extracellular bone matrix is commonly obtained by acid extraction and sterilization. Several authors have reported that adequate demineralization is achieved with a hydrochloric acid concentration of 0.6 M [26,27,28,29,30]. Nonetheless, after demineralization, many commercially available bone scaffolds exhibit alterations in their physical, chemical, biological, and mechanical properties, which impact clinical outcomes.

EDS yielded semiquantitative information about the inorganic composition of the scaffolds obtained with different concentrations of HCl. The scaffolds did not show well-defined peaks of calcium and phosphorus, thus explaining why the absence of calcium and phosphorus are indicators of adequate scaffold demineralization (Figure 4). The results also confirm the evidence obtained in Masson’s trichrome histological sections of 0.25 M and 0.5 M scaffolds; however, in the 0.125 M, 0.25 M, and 0.5 M scaffolds, a few points of mineralized matrix were observed with the same staining method. This mineralization was not detected by EDS because the detection limit for three-dimensional materials is 0.1 weight %; therefore, EDS cannot detect trace elements in powder form at concentrations less than 0.01 weight % [31]. In addition, other minerals that are important for bone homeostasis are found in lower proportions in bone tissue. These minerals include magnesium, which had a well-defined peak in all the scaffolds. Numerous minerals are found in a lower proportion in bone, and these minerals are essential for bone homeostasis. Magnesium is among these minerals, and it is important to mention that the magnesium peak remains well defined in the scaffolds. Magnesium degradation triggers calcium phosphate deposition [32,33], which is favorable for bone remodelling [34]. Studies have used various bioactive elements (e.g., magnesium and zinc) and incorporated them into scaffold surfaces to promote osteointegration and favor bone healing [35,36,37,38,39]. Reports have also shown that human periosteal cells exposed to magnesium exhibit increased expression of osteogenic genes and proteins [40].

We successfully demineralized the cancellous bone with HCl at 3 different lower concentrations compared with concentrations previously established by other authors [26,27,28], and demineralization was performed in less than 24 h by using HCl in combination with EDTA to eliminate the two different phases of calcium phosphates. The scaffolds also included magnesium, and its presence can have a favorable effect on mineralization and osteogenesis as mentioned above.

The Raman spectroscopic analysis of our samples clearly shows most characteristic Raman bands of bone tissues reported in the literature (Figure 5) [41]. For example, all our Raman spectra exhibit evident bands at 1669 cm^−1^ for stretching (str.) of amide I (C=O), 1456 cm^−1^ for deformations (def.) of CH_2_ and CH_3_, 1279 cm^−1^ for amide III (NH_2_ def.), 1256 cm^−1^ for amide III (C-N str.), 1101 cm^−1^ for (NCH def. of proline), 1000 cm^−1^ for phenylalanine, 932 cm^−1^ for the C-C str. of protein backbone, 859 cm^−1^ for the C-C str. of proline ring, and 828 cm^−1^ for the C-C str. of protein backbone. These samples show narrow bands at 1669 cm^−1^, 1456, 1279 cm^−1^, 1256 cm^−1^, and 1000 cm^−1^ and less defined bands at 1101 cm^−1^, 932 cm^−1^, 859 cm^−1^, and 828 cm^−1^ (Figure 5).

Conformational changes in type I collagen are associated with the relative intensities of bands at 859 cm^−1^, 932 cm^−1^, 1256 and 1279 cm^−1^ (amide III), and 1669 cm^−1^ (amide I). Proline constitutes approximately 10% of the total amino acids in collagen protein. The three parallel polypeptide strands are stabilized for polyproline II-type (PPII); therefore, the pyrrolidine ring directly affects the triple helix structure [42]. The protein backbone is a polypeptide chain responsible for determining the fold of a protein and provides its overall shape or tertiary structure [41]. Amide III and I bands are uniquely useful for collagen conformational analysis because they provide information on conformational changes of collagen molecules and secondary structures, such as α-helices, β-sheets, β-turns, and random coils [29,30]. This evidence is consistent with the fact that these bands are associated with the random/unordered structure of type I collagen protein [11], and our results provide sufficient evidence that the main component of the 0.125 M, 0.25 M, and 0.5 M scaffolds is type I collagen, which exhibits different degrees of hydrolysis. However, we did not perform any tests to determine the molecular weight of hydrolyzed type I collagen.

The FT-IR analysis of prepared samples as well as the type I hydrolyzed collagen were performed complementary to Raman spectroscopy, to obtain further information about the structural composition of samples and hydrolyzed collagen after treatments. FT-IR spectra are shown in Figure 6. The reference spectrum of type I hydrolyzed collagen shows characteristic protein bands; for example, a broad band at 3305 cm^−1^, which is associated with N-H str. and hydrogen bands present in collagen (amide A), and band at 3081 cm^−1^, associated to C-H stretching (amide B). In addition, the fingerprint region shows the following bands: 1654 cm^−1^, which is associated with the str. vibrations of the carbonyl group together with the polypeptide backbone (amide I) [41,42]; 1549 cm^−1^, which is associated with a shoulder; 1270 cm^−1^ of NH bending vibrations and C-N stretching (amide II); 1453 cm^−1^, which is associated with the pyridine ring that stabilizes the collagen triple helix by intrachain H+ bridges; 1239 cm^−1^ which is associated with C-H stretching; 1084 cm^−1^, which is associated with hydroxyproline linking the prolyl hydroxyl groups and the main-chain carbonyl groups; and 1041 cm^−1^, which is associated with proline side chains that allow torsion of the collagen triple helix [43].

It has been previously reported that the bands at 1030 cm^−1^, 1059 cm^−1^, and 1083 cm^−1^ arise from the C—OH stretching vibrations of glycoproteins, including osteonectin, alkaline phosphatase, and proteins with the RGD motif, such as fibronectin, fibrinogen, osteopontin, vitronectin and bone sialoprotein (BMP), in the extracellular bone matrix [44,45,46]. These results reveal that the characteristic bands for type hydrolyzed I collagen, and these findings are consistent with those previously reported [41,42,43,44,45,46].

The mechanical properties quantified in this work exhibit variability among the three different groups, and significant heterogeneity was noted among the different regions evaluated on each sample. This finding was expected as similar information was reported elsewhere for tensile tests on similar demineralized bone samples [46,47,48].

The mechanical properties of the scaffolds, particularly the Young’s modulus, represent essential characteristics and native properties of tissues and organs that must be restored in fabricated models for better compatibility based on a biomimetic tissue engineering approach. Cells can sense the microenvironment’s mechanical properties, and cell behavior is affected by mechanotransduction of the substrate. The results obtained from indentation are presented in Table 1.

The values of the elastic modulus obtained in the demineralized bone scaffolds measured here seem inferior to those reported in several other reports where values of up to GPa were found [41,42,43,44,45,46,47,48]. However, the findings are significantly consistent with another article where Young’s moduli of 200–300 MPa were measured using uniaxial tensile stress [29].

In particular, the properties of scaffolds derived from the bone extracellular matrix are different because they depend to a great extent on their composition, including the amount of hydrolyzed collagen, the type of collagen that constitutes it, and the porosity [30,37,49,50,51].

Thus, bone scaffolds in which the mineral phase of the bone and lipids have been removed, which have a low concentration of proteoglycans and different degrees of a breakdown of the collagen fiber bonds by HCl, lead to a reduction in biomechanical stiffness [28,29]. In this sense, demineralized bone xenografts are considered to have a lower modulus of elasticity [36].

On the other hand, the significant standard deviation found in our mechanical results confirms the great heterogeneity of demineralized bones. Therefore, this variation was expected for biological tissue samples because variations could occur in the collagen arrangement fibers in the trabeculae, the directionality of collagen fibers at the measured site, and the macropores present in the trabeculae the structure. Furthermore, data reported in [39] demonstrate that this biological material also relaxes because it exhibits viscoelastic properties that were not characterized here. This feature can explain further the significant variability. Thus, our results were obtained using the same measurement conditions on all scaffolds, thus providing sufficient evidence for comparing the different tested materials. The Young’s modulus values for the 0.5 M (*p* = 0.7) and 0.25 M (*p* = 0.5) scaffolds were like that of the commercial scaffolds Cancellous Bone Chips Biograft^®^ (Table 1). In the case of the 0.125 M (*p* = 0.003) scaffolds, the Young’s modulus was greater than that of the commercial scaffolds. In general, the elastic modulus seems to be inversely proportional to the hydrochloric acid concentration. After the demineralization process, the basic microstructure of the bone matrix was preserved. We attribute these results to the methods used to obtain the commercial scaffolds Cancellous Bone Chips Biograft^®^. The scaffolds are composed of 70% inorganic salts and 30% organic matrix, and the organic portion consists of greater than 90% collagen. The commercial scaffold preparation method is not known for its precision, and the assessment of the microstructure of the scaffold was beyond the scope of this investigation.

The swelling rate was relatively rapid in all groups, greater than 49% after 5 h of immersion in PBS. The highest percentage of swelling at 168 h was in the scaffolds 0.5 M obtained with the highest concentration of HCl; they exhibited a 186% ± 27.6 swelling ** *p* < 0.0001, while the 0.25 M and 0.125 M scaffolds 78.42% ± 19.5, and 86.45% ± 17.8 swelling of their weight respectively * *p* < 0.05 compared with commercial scaffolds Cancellous Bone Chips Biograft^®^ (Figure 7).

The swelling capacity is an essential property of the scaffolds and is also a factor determining the usability of in vivo applications. This property can be controlled in polymer manufacturing based on the type and degree of cross-linkers. However, cross-linkers are not recommended for scaffolds obtained from organs and tissues because they can cause immunological and inflammatory responses [19].

We expected the obtained these results because mineralized bone properties are different from scaffold properties. In general, after demineralization with reagents, we mainly preserved the extracellular matrix composed of proteins. However, extraction, solubilization, and denaturalization of the proteins are associated with the mineral phase [52,53,54], which includes type I collagen and proteoglycans [55,56,57].

The changes in the swelling capacity between the different groups of scaffolds can be attributed to the degree of hydrolysis of collagen protein. As mentioned above, hydrolyzed collagen has different properties than nonhydrolyzed collagen, including better water uptake capacity [58]. This feature could explain why the 0.5 M scaffolds exposed to higher HCl concentrations exhibited greater swelling capacity than the scaffold groups exposed to lower concentrations of 0.125 and 0.25 M HCL. The swelling capacity can also be linked to the mechanical characterization described above. A lower overall stiffness may be explained by larger pores in porous materials, hence a greater degradability by a buffer that may permeate more freely inside a softer material.

The data strongly suggests that the 0.125 M, 0.25 M, and 0.5 M scaffolds exhibited different weight losses according to the HCl concentration used during processing. The 0.5 M scaffolds lost 80.71% ± 3.7 of their weight in 7 days, while the 0.25 M and 0.125 M scaffolds lost 78.42% ± 9.3, and 86.45% ± 17.2 of their weight respectively in 28 days (Figure 8), HCl is capable of damaging the structure of type I collagen and decreased the amount of small leucine-rich proteoglycans (SLRPs) associated with type I collagen [26,27,28]. Thus, the scaffold 0.5 M exhibited a more fragile structure, and maybe some of its components were lost [27,33] because we expected a shorter degradation time than to the other scaffolds.

We used HEPES buffer to maintain enzyme activity between buffer changes; therefore, lysozyme and collagenase enzymes were consistently active to simulate conditions in vivo [54]. We only used two enzymes to simulate in vivo conditions. Given that the main components of the scaffolds are type I collagen and proteoglycans (to a lesser extent), we opted to use specific enzymes.

The collagenase enzyme causes degradation of the intact collagen triple helix [52]. The aggregates of collagen type I molecules are degraded starting from the exterior [53].

Type I collagenase is essential for the tissue remodeling process because it cleaves the native helix of fibrillar collagen I to make it degradable by another matrix metallopeptidase (MMP) [54]. Lysozyme is the most abundant enzyme in plasma, and its high affinity for proteoglycans has been observed [55,56]. Thus, the results obtained in the scaffold degradation times can vary under in vivo conditions. In the in vitro cytocompatibility tests, the macrostructure of the scaffolds was conserved for more than 28 days in the presence of hWJ-MSCs. These cells secrete paracrine factors that upregulate MMP expression [57] and have a proteolytic effect [58]. Furthermore, the enzyme concentrations approximated the physiological conditions. Nonetheless, we did not consider the buffering effect of other plasma components or the enzymatic effect acceleration due to the constant stirring speed.

### 3.2. hWJ-MSCs Isolation, Expansion, and Phenotypic Profile

The hWJ-MSCs at passage 0 exhibited a homogenous population and spindle-shaped morphology. The hWJ-MSCs were maintained in culture until they reached 70–80% confluence because prolonged culture time can cause modifications in gene expression, stem cell markers, and differentiation capacity [59,60].

Flow cytometry revealed that hWJ-MSCs at passage 3 were strongly positive for mesenchymal markers CD73, CD90, and CD105 and negative for hematopoietic stem cell markers CD34, CD45, CD11b, CD19, and human leukocyte antigens HLA-DR and HLA-ABC (Figure 9). This phenotype is consistent with the International Society for Cellular Therapy (ISCT) criteria established for the characteristics of the culture of MSCs [59,60].

MSCs can be divided into adult or fetal/perinatal MSCs, and hWJ-MSCs are considered perinatal-derived MSCs. Perinatal-derived MSCs represent the ideal option for therapeutic use because they have a low immunogenic phenotype [61] and immunomodulatory capacity and are easily expanded in culture. In addition, Perinatal-derived MSCs share many characteristics with adult MSCs, such as the potential for proliferation and differentiation in different cell types. However, these cells do not have the disadvantages of adult MSCs, such as decreased proliferative and decreased anti-inflammatory capacity [23]. Additionally, perinatal-derived MSCs can retain various embryonic stem cells (ESCs) characteristics but do not form teratomas [24]. Moreover, research on and applications of hWJ-MSCs in regenerative medicine are not associated with ethical or legal concerns.

### 3.3. In Vitro Assays

The scaffolds showed positive immunostaining for RGD motifs; many adhesion proteins in ECM are positive to CD51 and recognize RGD motifs.

The immunostaining data strongly suggested that 0.125 M and 0.25 M HCl are the optimal concentrations to expose RGD motifs 83.39% ± 4.65 and 78.73% ± 4.01, respectively, and there are no significant differences between both groups. In contrast, the scaffolds obtained with 0.5 M HCl had a lesser CD51 expression 22.69% ± 0.92, but the 0.125 M and 0.25 M scaffolds present significant differences * *p* < 0.0001 compared to bovine bone. The RGD motifs are hidden in the triple helix structure in the extracellular matrix. The results obtained are attributed to the notion that most RGD motifs were exposed at one specific point; a lower concentration of HCl cannot expose all RGD motifs but were subsequently degraded due to the highest HCl concentration. As a control for immunostaining, we used unprocessed bone 59.78% ± 1.64 (Figure 10).

Type I collagen denaturation has been reported to favor the exposure of RGD motifs hidden in the triple helix structure given that these sequences are exposed when the structure is denatured [21]. The RGD motif is the major integrin-binding site and comprises the amino acids arginine, glycine, and aspartic acid. The RGD motif is present in bone ECM proteins, such as fibronectin, vitronectin, osteopontin, osteonectin, vitronectin, and bone sialoprotein [62,63,64]. Integrin binding to RGD motifs regulates cell adhesion, proliferation, and differentiation [22,65,66].

We used an anti-CD51/ITGAV antibody to detect RGD motifs in the scaffolds; the presence of RGD motifs indicates that they may be used successfully for bone formation because they probably allow the hWJ-MSCs adhesion, proliferation, and differentiation.

The number of living cells in contact with the scaffolds was next analyzed (Figure 11A). On day 1 of culture, viable and attached cells were observed in all the scaffolds; however, more cells were in the commercial scaffold. On day 3 of culture, the highest number of living cells (green fluorescence: calcein-AM) corresponded to the 0.25 M scaffolds at 3, 7, 14, and 28 days of culture 17 ± 4.05, 255 ± 7.02, 875 ± 13.4, and 806 ± 8.6 respectively ** *p* < 0.001 compared with 0.125 M, 0.5 M and commercial scaffolds. While in the 0.125 M and 0.5 M scaffolds at 7, 14, and 28 days of culture, there were more viable cells than in the commercial scaffolds compared with commercial scaffolds * *p* < 0.05 (Figure 11B).

Furthermore, in the scaffolds 0.125 M and 0.25 M, we noticed a cellular orientation in a preferential direction; this may be because collagen fibers and RGD motifs provide topographical cell guidance and produce aligned cellular orientation.

However, we did not observe dead cells on the scaffolds (red fluorescence—Ethidium homodimer), given that the hWJ-MSCs exhibited anoikis, a form of death in anchorage-dependent cells as their viability depends on their adhesion. Previously we seeded monolayer hWJ-MSCs to establish them as live cell controls and for dead cells, and they were treated with ethanol (Figure 12).

A fusiform shape of the cells was observed in all scaffolds from day 7 of culture, attributed to the integrin-mediated firm cell adhesion phase [57]. This finding can also be attributed to the cell adhesion phase in which the cells were in the first stage of contact, which is a low-affinity stage.

Prior to cell culture, we suspended the hWJ-MSCs in the DMEM-F12 medium. The cell suspension was held inside the scaffolds by capillary force, which allowed cells to establish low-affinity contacts followed by integrin-mediated firm adhesion. Notably, the scaffold represented a change in the cellular microenvironment because the hWJ-MSCs were previously seeded in culture flasks under more rigid conditions than those of the scaffolds. Cell guidance can be provided by external cues, such as the chemical composition, substrate rigidity, and substrate topography of the scaffolds [58]. Sufficient evidence has shown that ordered micropatterned structures and disordered nanopatterned structures occur, in which the physical microenvironment and the cell–material interface can guide cell fate and regulate these behaviors [66]. We observed cell orientation and directionality in the 0.25 M scaffolds. This finding is significant and can be attributed to the presence of RGD motifs. RGD motifs favor cellular adhesion to scaffolds, whereas adhesion can also be influenced by the materials’ surface nanotopography [64]. The nanotopography of collagen fibers alignment could be observed in histological sections via Masson’s trichrome staining. This finding is significant and can be attributed to the presence of RGD motifs. RGD motifs favor cellular adhesion to scaffolds, whereas adhesion can also be influenced by the materials’ superficial nanotopography [64]. We were able to associate the presence of RGD motifs in the 0.125 M and 0.25 M scaffolds based on the observed cell adhesion of hWJ-MSCs [64,65,66].

We used calcium phosphate deposition as an indicator of cell differentiation in scaffolds using von Kossa staining (Figure 13). Recently, different approaches have been performed to obtain bone fillers. A protein extract of bone extracellular matrix with polycaprolactone, [18] hydroxyapatite granules obtained from chicken eggshell [20], and bovine bone ECM granules [19] have been tested. It should be noted that these bone fillers, although not cytotoxic, their capacity for osteoinduction, osteoconduction, and osteogenesis in vitro is minimal. On the other hand, since they are granules, their ability to support bone is limited. However, in the present study, we demonstrated that the 0.125 M and 0.25 M scaffolds showed a substantial deposition of calcium phosphate after 28 days of culture in the absence of mineralization-inducing medium 10.85% ± 1.3 and 71.38% ± 19.16 respectively * *p* < 0.0001 compared with the bovine bone; although, the mineralization rate observed in the 0.25 M scaffolds overcame the deposition of calcium in 0.125 M scaffolds. At 14 days, the scaffolds exhibited an ochre and reddish coloration that indicated calcium phosphate and osteoid deposits, respectively. At 28 days, the scaffolds were very brittle; therefore, it was challenging to obtain histological sections. The 0.125 M scaffolds only exhibited weak ochre staining in some areas on the periphery of the trabeculae at 14 days and ochre staining on the periphery of the entire trabeculae and some sites with reddish coloration at 28 days 7.39% ± 4.36 * *p* < 0.0001 compared with the bovine bone; in contrast, the 0.5 M scaffolds exhibit yellow staining at 14 days and an ochre staining extension on the trabecular periphery at 28 days 0.72% ± 0.42. Evident red staining was observed in all the remaining tissue. The trabecular structure changes and tissue degradation in this last group should be highlighted. We observed that the scaffold consistency was stickier when generating sections, which prevented the trabeculae from being separated and extended on the coverslip.

The 0.25 M scaffolds had a favorable microenvironment with osteoconductive and osteoinductive capacities, which are indispensable for promoting tissue-specific remodeling and act as a template for the repair and functional reconstruction of three-dimensional bone tissue. In the in vitro studies performed in the present work, the osteogenic differentiation potential of hWJ-MSCs was evident in the absence of a mineralization-inducing medium. Furthermore, the osteogenic differentiation potential of perinatal MSCs was consistent with that these cells exhibit superior osteogenic therapeutic potential compared with adult MSC [23]. Furthermore, Perinatal MSCs have increased expression levels among all 16 osteogenic genes [64], higher levels of calcium deposits, and increased alkaline phosphatase activity [63]. These findings correlate with our findings, given that von Kossa staining is a highly reproducible and widely used technique to determine the presence of calcium phosphate deposits and osteoids in bone tissue engineering despite the difficulty of generating fine sections of scaffolds and their evident fragility.

In this study, scaffolds processed with 0.25 M HCl exhibited osteoinductive and osteoconductive capacity because they could induce cell adhesion and showed broad expression of RGD motifs. Integrins act as cell receptors via the RGD motif. These receptors’ fundamental characteristics are based on their ability to transduce signals into the cell and modulate signal cascades induced by different growth factors. In addition, hWJ-MSCs exhibited 14 times greater viability than commercial scaffold and osteogenic potential because they favor the calcium phosphate deposits in the scaffolds without a mineralization-inducing medium.

## 4. Conclusions

Several authors have reported that the ideal concentration of HCL to obtain a demineralized bone matrix is 0.5 M. Nonetheless, in the present study, we show that scaffolds obtained with 0.25 M HCl present osteoconductive capacity hWJ -MSC adhered throughout their 3D scaffolds. On the other hand, the demineralized bone matrix obtained in the present study also presents osteogenic potential since calcium phosphate was deposited on the scaffolds without a mineralizing inductor medium.

The scaffolds obtained under the conditions already described are easy to obtain. Furthermore, the scaffold conditions are repeatable concerning its porosity, mechanical strength, composition, cell growth capacity, and cytocompatibility, making it a promising scaffold for bone tissue engineering.

In future in vivo trials, we will assess the response to scaffold implantation and bone formation.

## Figures and Tables

**Figure 1 polymers-13-04390-f001:**
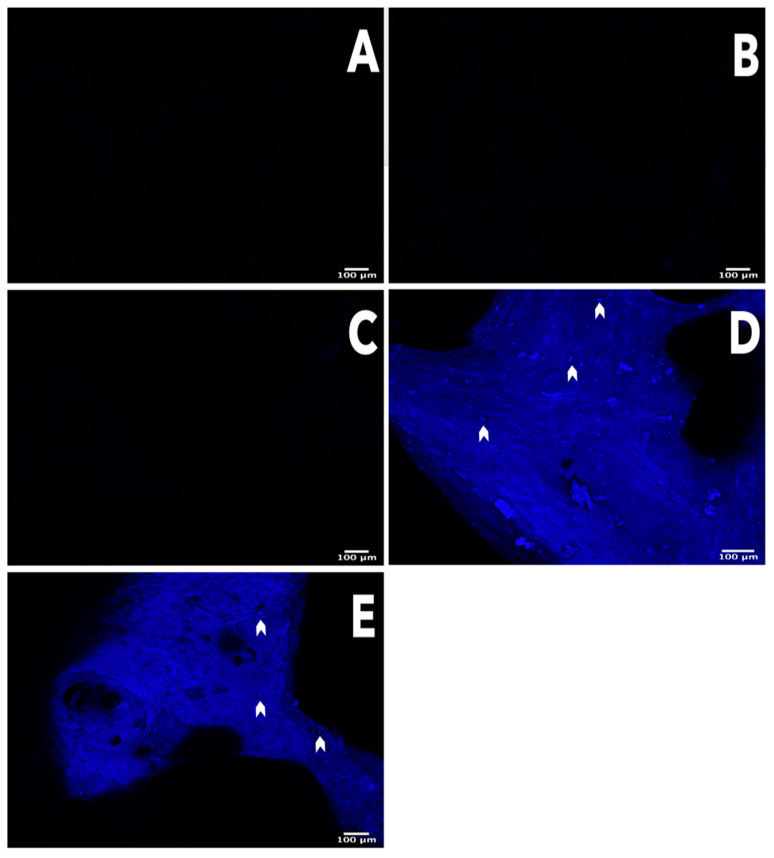
Confirmation of the decellularization process. Nuclear content was tested with 4′,6-diamidino-2-phenylindole (DAPI) staining. Confocal microscope photomicrographs of histologically stained sections with DAPI. Bright spots (white arrowheads) indicating nuclei can be seen before decellularization (**D**) and in commercial scaffolds (**E**) Nuclei are totally absent after decellularization in 0.125 M (**A**), 0.25 M (**B**), and 0.5 M (**C**) scaffolds. 40× scale bar = 100 μm.

**Figure 2 polymers-13-04390-f002:**
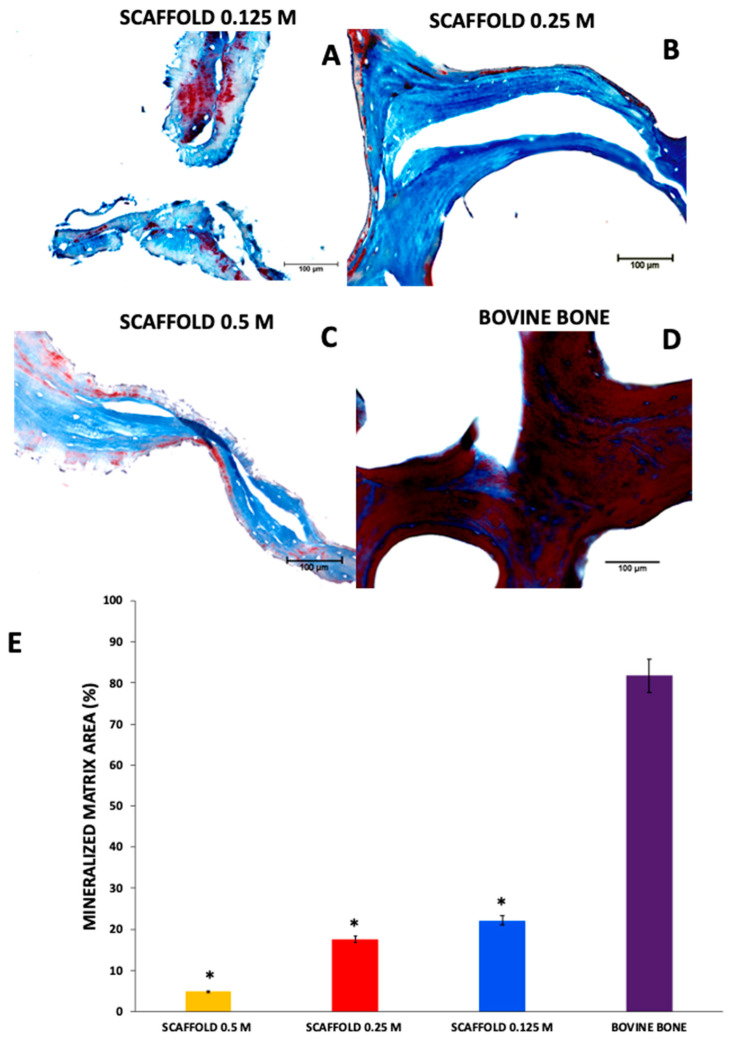
Histological analysis of the scaffolds and bovine bone. Optical microscope photomicrographs of histologically stained sections with Masson’s trichrome collagen fibers (blue) and mineralized matrix (red). The 0.125 M (**A**), 0.25 M (**B**), and 0.5 M (**C**) scaffolds were composed of collagen (blue). Nevertheless, in the 0.125 M scaffold, some zones indicate the presence of mineralization (red). As we expected, the bovine bone without the demineralization process (**D**) is composed of collagen (blue) and a mineralized matrix (red). The mineralized matrix area evidenced in Masson trichrome staining was calculated by Image-pro software (**E**). * *p* < 0.0001 compared with bovine bone without the demineralization process. Mean + SD *n* > 3. 40×, scale bar = 100 μm.

**Figure 3 polymers-13-04390-f003:**
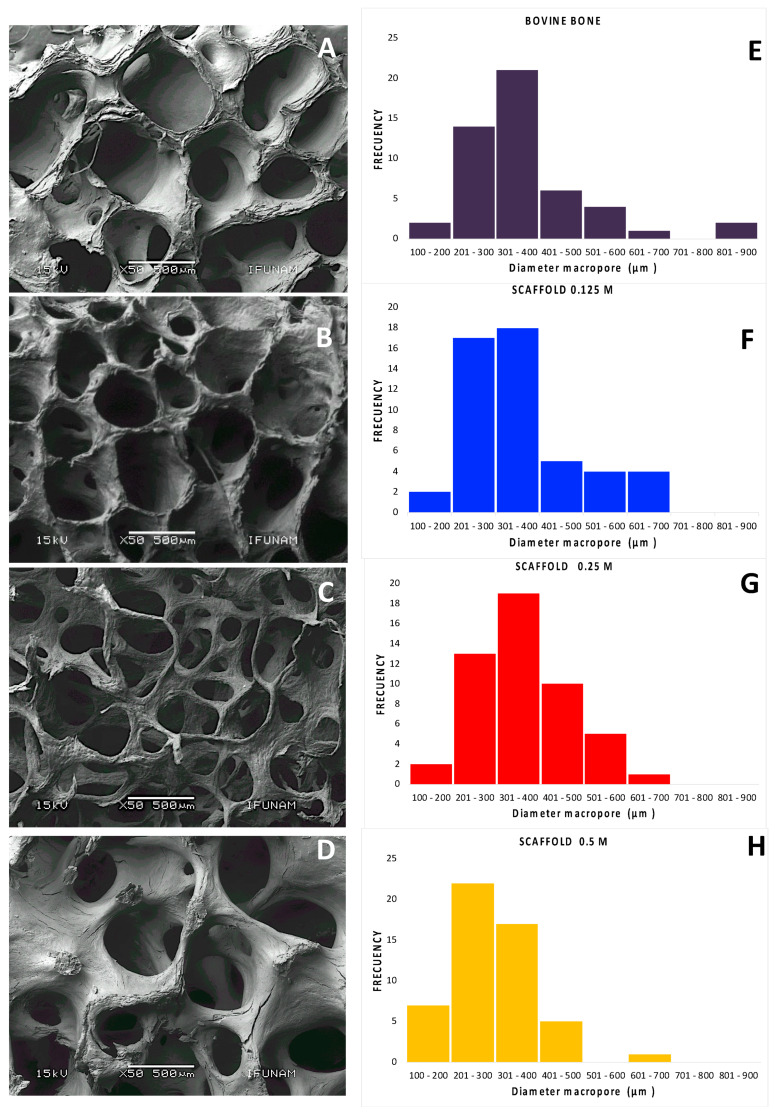
A-D SEM photomicrographs by the backscattered electron technique. 0.125 M (**A**), 0.25 M (**B**), 0.5 M (**C**), and bovine bone (**D**). **E**–**H** Histogram of the pore size distribution determined by image J software. Diameter macropore is indicated as the equivalent larger macropores diameter observed in **A**–**D**, 0.125 M scaffolds were 356.69 μm ± 125.36 (**E**), 0.25 M scaffolds were 352.82 μm ± 96.65 (**F**), 0.5 M scaffolds were 300.08 μm ± 94.96 (**G**), and the bovine bones without processing as control were 372 μm ± 141.6 (**H**). Mag.: 50×, scale bar = 500 μm.

**Figure 4 polymers-13-04390-f004:**
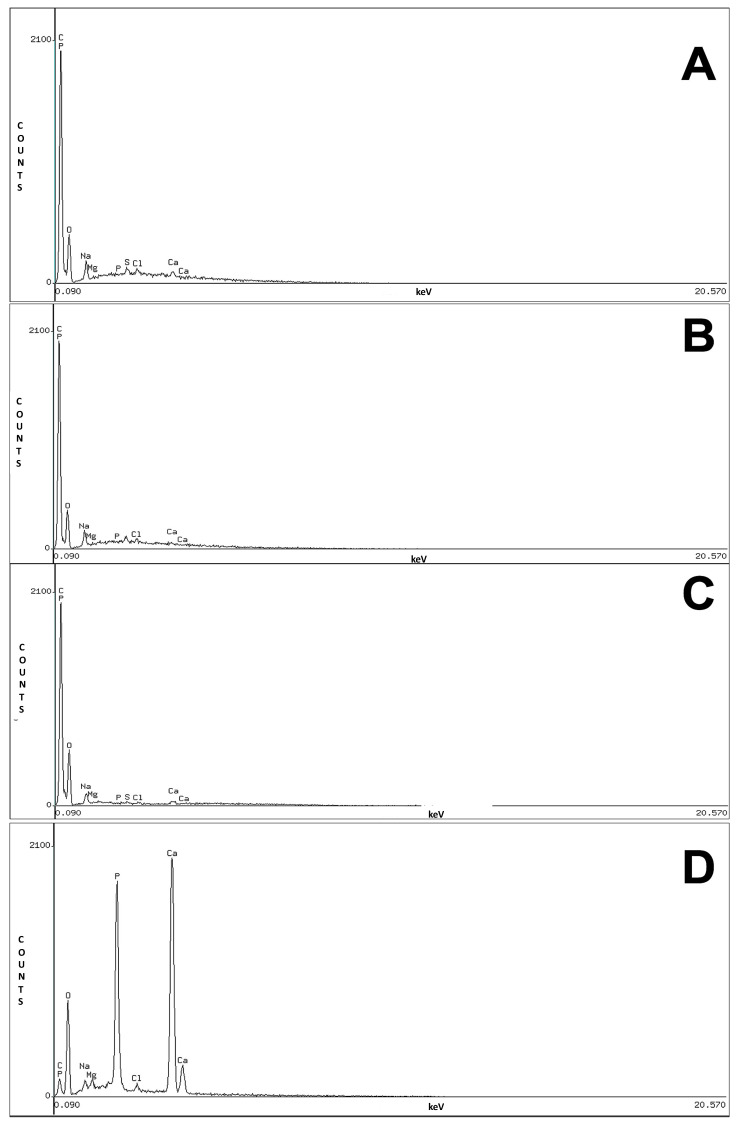
Chemical analysis by EDS indicated that the scaffolds did not contain calcium or phosphorus and therefore were demineralized. In contrast, mineralized bovine bone had a high calcium and phosphorus content. 0.125 M (**A**), 0.25 M (**B**), 0.5 M scaffolds (**C**), and (**D**) bovine bone.

**Figure 5 polymers-13-04390-f005:**
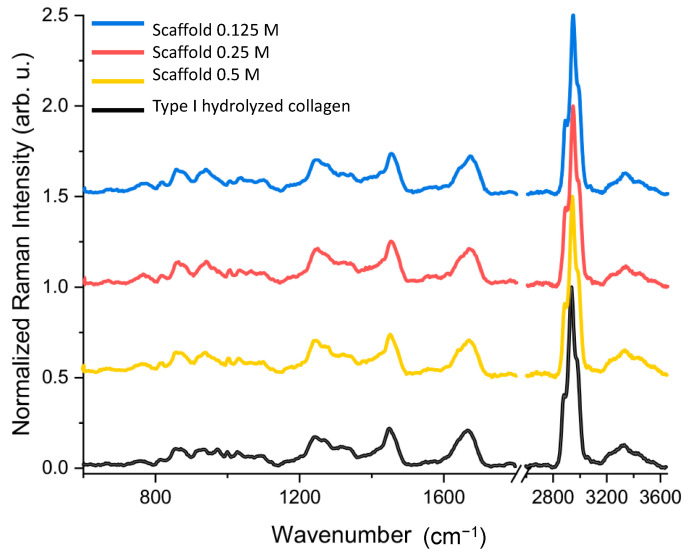
Raman spectra of previously prepared scaffolds with different treatments: blue (0.125 M), red (0.25 M) and yellow (0.5 M), as well as the type I hydrolyzed collagen as reference (dark). Representative Raman spectra, *n* > 3.

**Figure 6 polymers-13-04390-f006:**
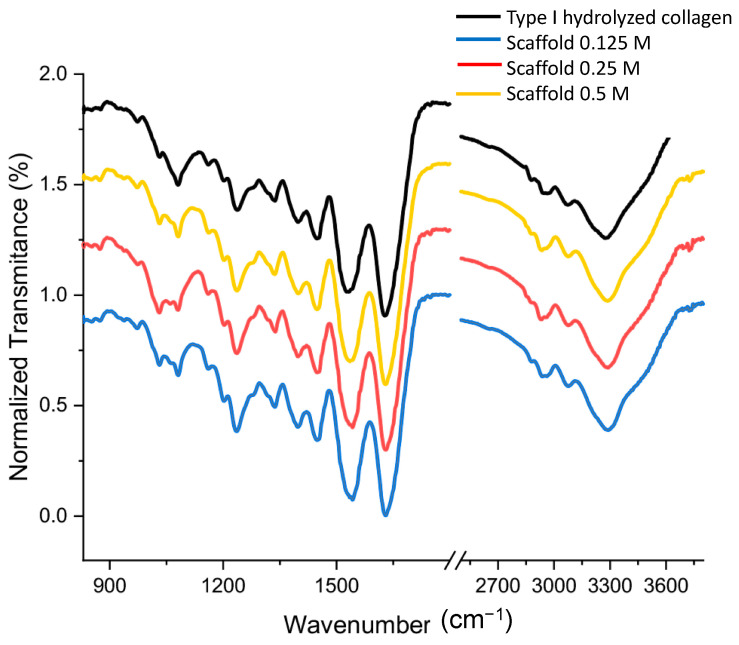
FT-IR spectra of prepared samples with different treatments: blue (0.125 M), red (0.25 M) and yellow (0.5 M), as well as the type I hydrolyzed collagen (dark). Representative FT-IR spectra *n* > 3.

**Figure 7 polymers-13-04390-f007:**
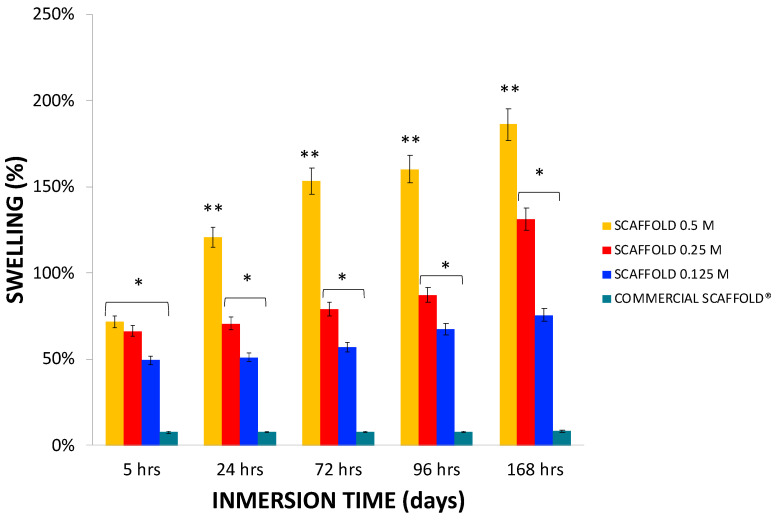
The swelling behavior of the scaffolds was determined gravimetrically by incubating scaffolds with PBS at 37 °C (pH 7.4) at different time intervals. The swelling of the 0.125 M and 0.25 M scaffolds was greater than that of the commercial scaffolds at 5, 24, 72, 96, and 168 h * *p* < 0.05, while the swelling of the 0.5 M scaffolds was greater than that of the 0.125 M, 0.25 M and commercial scaffolds at 24, 72, 96, and 168 h ** *p* < 0.0001. Mean ± SD *n* > 3.

**Figure 8 polymers-13-04390-f008:**
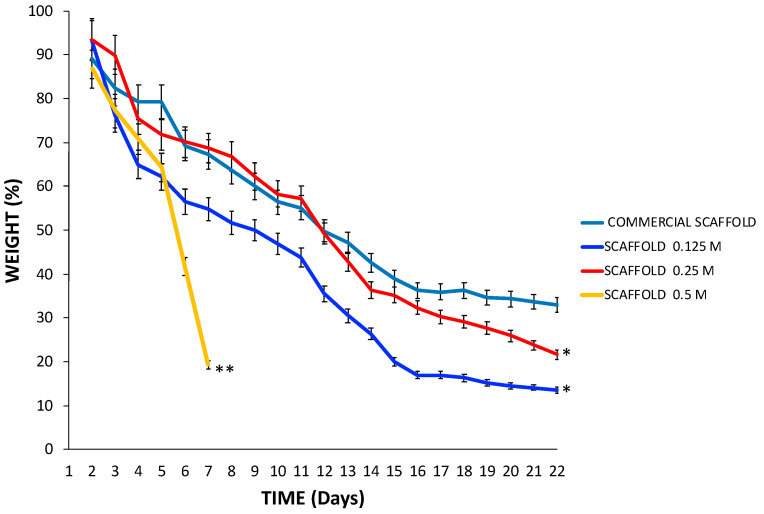
In vitro degradation rate was determined gravimetrically by treating scaffolds in a degradation buffer at 37 °C at continuously stirring. The 0.5 M scaffold, processed at the highest HCl concentration, lost more weight than 0.125 M, 0.25 M, and commercial scaffolds at 7 days ** *p* <0.0001. After this time, the 0.5 M scaffold dissolved in the degradation buffer, and it was impossible to evaluate it. The 0.125 M scaffolds, 0.25 M, lost more weight than commercial scaffolds at 28 days. * *p* < 0.05. Mean ± SD *n* > 3.

**Figure 9 polymers-13-04390-f009:**
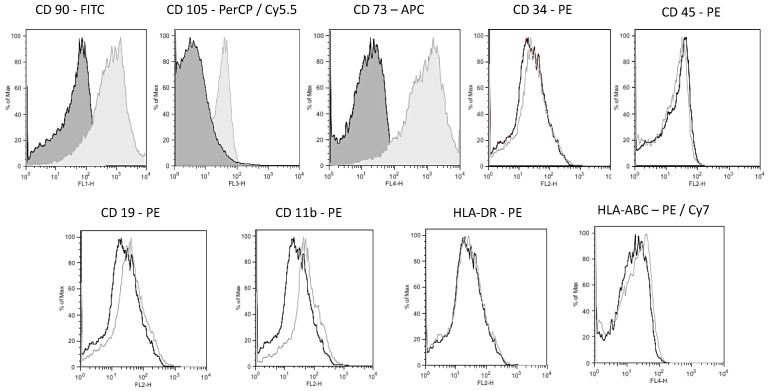
Flow cytometry histograms of hWJ-MSCs. Flow cytometry revealed that hWJ-MSCs at passage 3 were strong positive for mesenchymal markers CD73, CD90, CD105, negative for hematopoietic stem cell markers CD34, CD45, CD11b, CD19, and human leukocyte antigens HLA-DR and HLA-ABC in agreement with International Society for Cellular Therapy (ISCT) established minimal criteria for MSC culture characteristics.

**Figure 10 polymers-13-04390-f010:**
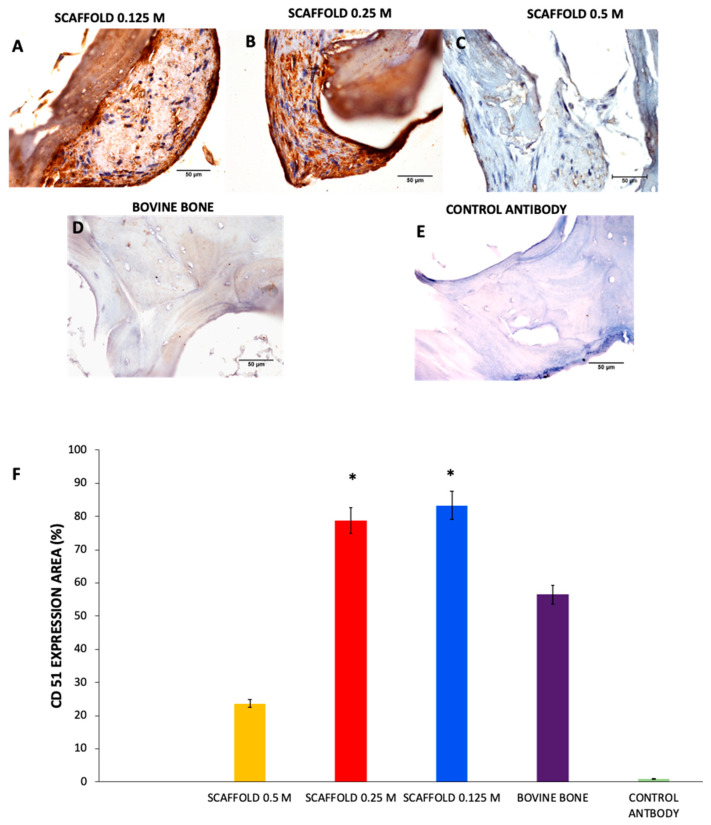
Immunohistochemical analysis in histological sections of the scaffolds and bovine bone immunolabeled with CD51 (brown). Scaffolds were positive to CD51 antibody staining in the 0.125 M (**A**), 0.25 M (**B**), and 0.5 M (**C**). As controls, we used bovine bone without decellularization process (**D**) and the bone without primary antibody (**E**). The CD51 expression area quantified was calculated by Image-pro software (**F**). The 0.125 M and 0.25 M scaffolds present significant differences compared to bovine bone and 0.5 M scaffold. * *p* < 0.0001, Mean ± SD *n* > 3. 40×, scale bar = 50 μm.

**Figure 11 polymers-13-04390-f011:**
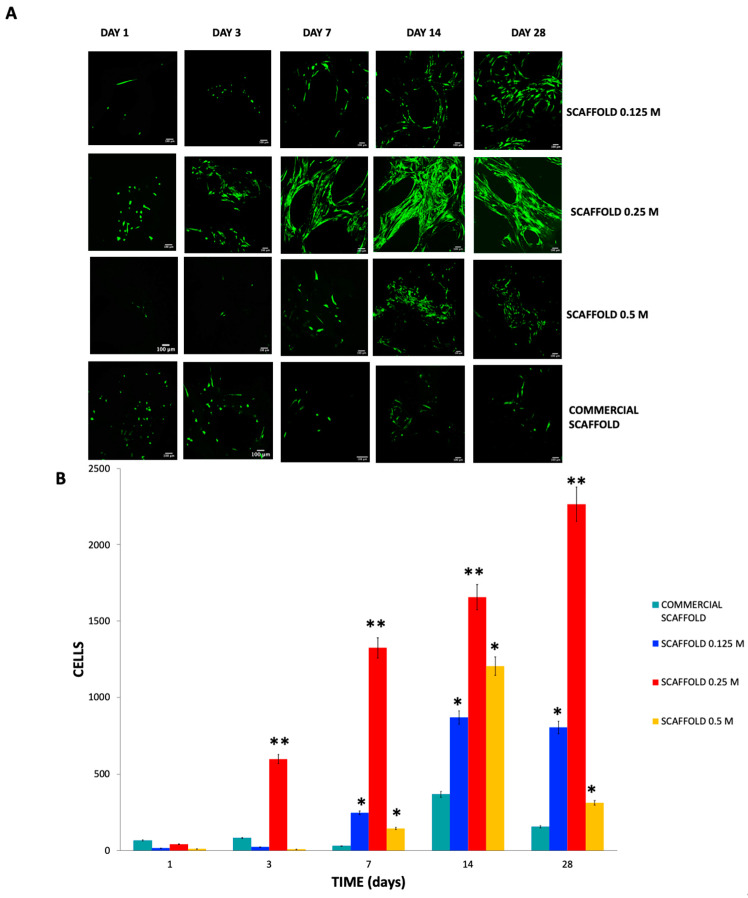
Confocal microscopy observations of in vitro viability of hWJ-MSCs by live/dead assay. Viable cells (green fluorescence-Calcein-AM), dead cells (red fluorescence-Ethidium homodimer), and cell nuclei (blue-DAPI). (**A**) Live/Dead assay of hWJ-MSCs cultured on the scaffolds. (**B**) Quantitative evaluation by ImageJ count cell of hWJ-MSCs cultured on the scaffolds. In the 0.25 M scaffolds at 3, 7, 14, and 28 days of culture, there are more viable cells than in the other groups ** *p* < 0.0001. In addition, in the 0.125 M and 0.5 M scaffolds at 7, 14, and 28 days of culture, there are more viable cells than in the commercial scaffolds compared with commercial scaffolds * *p* < 0.05. Mean ± SD *n* >3. 40×, scale bar = 100 μm.

**Figure 12 polymers-13-04390-f012:**
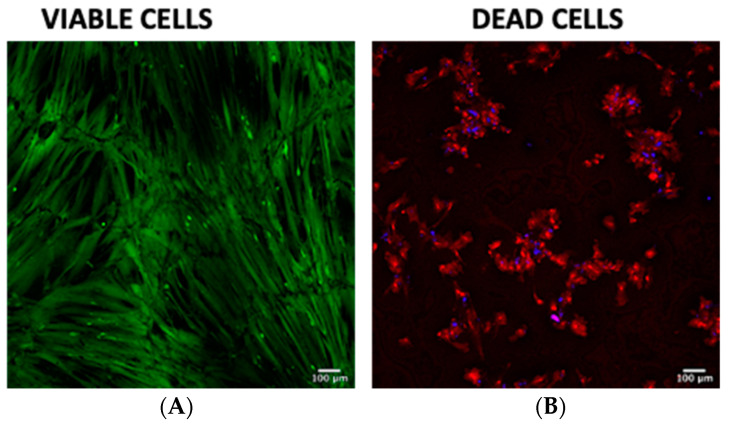
In vitro cell culture and viability of hWJ-MSCs. Confocal microscope microphotographs of hWJ-MSCs at 40× magnification. We seeded 10,000 hWJ-MSCs in monolayer on plain microscope slides. In both, we performed live/dead assay. We observed viable cells (green fluorescence—Calcein—AM) (**A**), and dead cells (red fluorescence—Ethidium homodimer) cell nuclei (blue—DAPI) (**B**).

**Figure 13 polymers-13-04390-f013:**
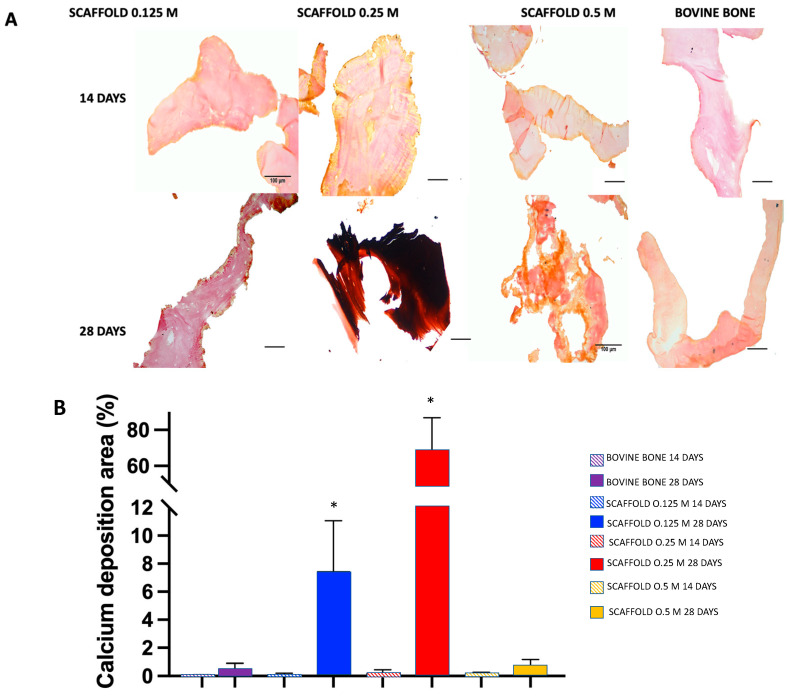
(**A**) Detection of mineral deposit visualized by von Kossa staining in histological sections: collagen (pink), osteoid (red), calcium (black). Optical microscope photomicrographs of histologically stained sections with von Kossa at 40× magnification. We used the visualization of calcium deposits, stained with the von Kossa technique, as an indicator of differentiation of the hWJ-MSCs cultured on the scaffolds without a mineralization-inducing medium. As a control, we used decalcified bovine bone. (**B**) The quantification of calcium deposit area in von Kossa staining calculated by Image-Pro software demonstrates that the 0.125 M and 0.25 M scaffolds showed a substantial deposit of calcium phosphate after 28 days of culture in the absence of mineralization-inducing medium compared to all groups * *p* < 0.0001. Mean + SD *n* > 3. 40×, scale bar = 100 μm.

**Table 1 polymers-13-04390-t001:** Young’s modulus of 0.125 M, 0.25 M, and 0.5 M scaffolds in wet conditions.

	Scaffold 0.125 M	Scaffold 0.25 M	Scaffold 0.5 M	Commercial Scaffold
Young’s modulus (kPa)	1122.65	385	231.2	282.6
Standard deviation	1352.69	476	237.8	470.5

## Data Availability

The data presented in this study are available on request from the corresponding author.

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
