# Peer review of "Three-Dimensional Porous Scaffolds Derived from Bovine Cancellous Bone Matrix Promote Osteoinduction, Osteoconduction, and Osteogenesis"

_polymers, 2021, doi:10.3390/polym13244390_

Round 1

Reviewer 1 Report

Paper titled (Three-dimensional porous scaffolds derived from bovine cancellous bone matrix promote osteoinduction, osteoconduction, and osteogenesis.) by Alda et al. Overall, authors did a good job in this paper but the presentation of figures is poor in most cases either for quality and appearance that need enhancement or the correctness and lack of significance symbols. I find these items are mandatory and essential for drawing the conclusion. 

1- Fig 3, Fig 7 and Fig 8:  axis are not clear and has not high resolution.

2- All figures have no significance symbols, all significances should be referred to it in the figures. WHy we did it if not presented ??

3- Introduction: I feel it is long, ensure it is not more than 700 words

Some very small paragraphs are there, kindly join them.

4-Can authors analyze images in figure  1, 2 and 12?

5- Pleas image analysis for figure 10? 

6- Title: osteogenesis was tested in what? mention the cell

Author Response

Referee one

Please, note that the line numbering on page 1 goes from 1 to 46, and on page 2, it jumps to line 104. The line numbering format is something that we could not correct in the document despite entering the corresponding commands in the Word document. We apologize for this situation.

Paper titled (Three-dimensional porous scaffolds derived from bovine cancellous bone matrix promote osteoinduction, osteoconduction, and osteogenesis.) by Alda et al. Overall, authors did a good job in this paper, but the presentation of figures is poor in most cases either for quality and appearance that need enhancement or the correctness and lack of significance symbols. I find these items are mandatory and essential for drawing the conclusion.

1. Fig 3, Fig 7 and Fig 8: axis are not clear and has not high resolution.

R. Thank you for your comments regarding the figures. We have changed the figures, given them higher resolution, and placed some lines in light blue to give more clarity to the data.

2. All figures have no significance symbols, all significances should be referred to it in the figures. WHy we did it if not presented??

R. Thanks for your observation. We have placed the significance in figure 7 and 8, which are the figures in which it was calculated. We apologize for not having placed it. It was a mistake.

3. Introduction: I feel it is long, ensure it is not more than 700 words. Some very small paragraphs are there, kindly join them.

R. We have shortened the introduction according to your suggestion. Before, it had 742 words, and now has 626 (L31-L146).

We rewrite some of the paragraphs, joining some and separating others.

4. Can authors analyze images in figure 1, 2 and 12?

R. Thanks for your suggestion. We have placed an analysis of each image in the corresponding figure captions. In addition, in the text is the analysis of each figure.

In the caption of figure 1 (L496 - L501), we add that the decellularization process of the bovine bone was adequate, and therefore, no nuclei were observed.

At the bottom of figure 2 (L551 – L556), we mention the great contrast between A, B, and C with D concerning the presence of Ca++. Which shows that the descaling process was adequate.

In Figure 12 (L1281 – L1288), we describe that there are more cells in ECM sponges obtained from bovine bone than in non-decalcified cancellous bone.

5. Pleas image analysis for figure 10? 

R. We have placed the analysis for figure 10 in the figure caption (L1170 – L1176). Antibody against CD51 identifies the ligand of the RGD motif. The CD51 molecule is present in adhesion proteins such as laminin, fibronectin, and osteonectin. The fact that there is a high expression of this ligand in the scaffolds processed with 0.125 and 0.25 M HCl suggests that these scaffolds will allow greater cell adhesion.

6. Title: osteogenesis was tested in what? mention the cell

R. The cell line on which osteogenesis was tested is mentioned in the abstract (L26-27), and in the material and methods section (L327). hWJ-MSCs are stem cells obtained from human umbilical cords. Osteogenesis was tested by seeding Warthon gelatin stem cells on scaffolds without mineralizing medium against commercial scaffold. It was proved, with the von Kossa staining (fig. 13) (L1364 – L1373), that demineralized scaffolds began to mineralize 28 days after stem cells were seeded.

Reviewer 2 Report

The present study proposes a facile 3 step procedure to obtain 3D porous scaffolds from bovine bone, while maintaining its biocompatibility and ability to interact with mesenchymal stromal cells. The work is impressive, its strong point being the in-vivo tests. However, the general impression was that the extensive compositional and morphological evaluation preceding the in-vivo tests was rather unnecessary, given the fact that every result was already correlated with literature. Hence, the novelty of the scaffold preparation procedure is unclear and maybe the paper should focus solely on the biological evaluation of such materials.

Please find my comments, point by point:

  1. If the authors find it suitable, maybe the title could be shortened.
  2. Please change the topic of large sentences, add commas or divide them into smaller parts in order to be easier to follow and understand (e.g., L16-19;

Introduction section

  1. It is not clear if this type of procedures were previously performed on bovine cancellous bone;
  2. If similar studies are described in literature, what is the added information (novelty) of this study?
  3. Lines 95-99 are more suitable as conclusion or abstract and usually are not stated in the introduction;

Materials and methods

  1. L 129: please add the type of solution with 0.5M concentration “immersed in 0.5 M for 2 hours”
  2. L150-151, L190: please define “commercial scaffold “(brand, batch, composition, producer etc.)
  3. L158: Why using SEM for such small magnification (50 X)? The samples were previously coated with a conductive metal?
  4. L186: please define DMEM, since it is first mentioned here.
  5. L196, L210, L227: This statement is rather unnecessary, since the results were statistically evaluated, as described later.

Results and discussion:

  1. Figure 3: please check the type of SEM technique used; I strongly believe it is not backscattered electron detection.
  2. Figure 4: the images seem to be results of an EDS analysis, yet the description presents them as Wide-angle X-ray diffraction patterns.
  3. Figures 5 and 6: it is not clear why the authors chose to perform Raman and FTIR analysis on the same wavelength range; what is the added value in using both techniques?
  4. Figure 7: please mark the statistical significance of the obtained results for each sample (e.g., with *, ** or letters)

References: please cite only relevant sources, related with the subject of this paper; the reference are outdated, only ~15% of the cited references are from 2019-2020 period and none from 2021; please add recent development in the field (see: J. Mater. Chem. B, 2021,9, 6881-6894; Nanomaterials 2021, 11(9), 2289; Journal of International Dental and Medical Research 2021, 14(2), 623-628)

Editing errors in all superscripts from the measurement units and some subscripts from chemical formulas.

Author Response

Referee 2

Please, note that the line numbering on page 1 goes from 1 to 46, and on page 2, it jumps to line 104. The line numbering format is something that we could not correct in the document despite entering the corresponding commands in the Word document. We apologize for this situation. 

The present study proposes a facile 3 step procedure to obtain 3D porous scaffolds from bovine bone, while maintaining its biocompatibility and ability to interact with mesenchymal stromal cells. The work is impressive, its strong point being the in-vivo tests. However, the general impression was that the extensive compositional and morphological evaluation preceding the in-vivo tests was rather unnecessary, given the fact that every result was already correlated with literature. Hence, the novelty of the scaffold preparation procedure is unclear and maybe the paper should focus solely on the biological evaluation of such materials.

Please find my comments, point by point:

1. If the authors find it suitable, maybe the title could be shortened.

R. We appreciate your suggestion; however, we consider that the title describes what was done and what is written in the article.

2. Please change the topic of large sentences, add commas or divide them into smaller parts in order to be easier to follow and understand (e.g., L16-19;

R. We appreciate your extensive review of the article. As a result, we have rewritten the sentence found on lines 16 to 19. We have also corrected other sentences throughout the text of the article.

Introduction section

3. It is not clear if this type of procedures were previously performed on bovine cancellous bone;

R. Thanks for your suggestion. We have added some paragraphs on the lines 39 to 113, 135 to 140, and 141 to 146. In these paragraphs, we have clarified the article's novelty, and we have also described the purpose of the article to clearly show the originality of obtaining bone scaffolds.

4. If similar studies are described in literature, what is the added information (novelty) of this study?

R. We greatly appreciate your kind comments to improve the article. Although biological scaffolds have been obtained from bovine bone ECM, these have been obtained at higher concentrations of HCL, which can significantly affect their morphological composition and biological properties. On the other hand, we obtained an ECM scaffold with lower concentrations of HCl. Therefore, it had to be characterized both mechanically and biologically. Furthermore, the scaffold obtained in this study has osteogenic, osteoinductive, and osteoconductive capabilities, unlike other scaffolds. In this sense, we have added some paragraphs (L135 – L146) in the introduction to note the novelty of the scaffolding preparation.

5. Lines 95-99 are more suitable as conclusion or abstract and usually are not stated in the introduction.

R. These lines are modified according to your suggestion. The lines in the revised document are 143-146.

Materials and methods

6. L 129: please add the type of solution with 0.5M concentration “immersed in 0.5 M for 2 hours”

R. We made the changes to the line (L219 in the revised document).

7. L150-151, L190: please define “commercial scaffold “(brand, batch, composition, producer etc.)

R. We have placed the data of the commercial bone scaffold. We have done it on the lines (L280, L441, and L449 of the revised document)

8. L158: Why using SEM for such small magnification (50 X)? The samples were previously coated with a conductive metal?

R. In this case (line 251 of the revised document), SEM was used due to the depth of field it offers and which far exceeds light microscopy. For this reason, the pores of the samples are clearly visible. On the other hand, we used small magnifications to measure the pores of the scaffolds.

Yes, the samples were covered with a thin layer of gold (of the order of nanometers) by the sputtering technique.

9. L186: please define DMEM, since it is first mentioned here.

R. DMEM was defined in line 276 of the revised document.

10. L196, L210, L227: This statement is rather unnecessary, since the results were statistically evaluated, as described later.

R. We have eliminated the paragraphs that contained the mentions to the statistical analysis and in the lines L 286, L 310, and L 327 of the revised document.

Results and discussion:

11. Figure 3: please check the type of SEM technique used; I strongly believe it is not backscattered electron detection.

R. The samples were observed with backscattered electrons, but we covered them with a thin film of gold to avoid possible edge effects and to observe the pores more clearly.

12. Figure 4: the images seem to be results of an EDS analysis, yet the description presents them as Wide-angle X-ray diffraction patterns.

R. Thanks for your observation. Indeed, the images in Figure 4 were taken with SEM / EDS. Therefore, we have removed the phrase "wide-angle X-ray diffraction patterns" from the line L720, and L722

13. Figures 5 and 6: it is not clear why the authors chose to perform Raman and FTIR analysis on the same wavelength range; what is the added value in using both techniques?

R. In this work Raman and FT-IR analysis were included first as complementary techniques, to provide further detailed analysis on structural composition and protein content analysis. The main advantage of Raman spectroscopy was the capability to show us, with major detail, the protein content variations, and tentatively conformational changes along with each scaffold fabrication. On the other hand, besides the protein analysis, FT-IR spectroscopy allowed us to obtain further chemical structural information of the scaffold, for example, in the extracellular bone composition, with better resolved bands related to C-OH stretching vibrations, that are normally of week intensity in Raman spectroscopy. A small description of using both techniques was included in the FT-IR results section (L 868 – L872 of the revised document).

14. Figure 7: please mark the statistical significance of the obtained results for each sample (e.g., with *, ** or letters)

R. We have added asterisks to indicate significance in figure 7, and figure 8.

15. References: please cite only relevant sources, related with the subject of this paper; the reference are outdated, only ~15% of the cited references are from 2019-2020 period and none from 2021; please add recent development in the field (see: J. Mater. Chem. B, 2021,9, 6881-6894; Nanomaterials 2021, 11(9), 2289; Journal of International Dental and Medical Research 2021, 14(2), 623-628)

R. We have added a paragraph in the discussion section with the references you suggested to us (L1381-L1383 of the revised document). We have also decreased old references and added more recent ones.

16. Editing errors in all superscripts from the measurement units and some subscripts from chemical formulas.

R. We have corrected all superscripts and subscripts in the article text.

Round 2

Reviewer 1 Report

Thanks for authors for revising the manuscript however, I think some revisions are still mandatory & authors did not reply to the previous questions & the same problems in figures are persist + colored columns have no clear axis or good resolution to show writing.

1- Figures 1,2 and 12, were not analyzed or quantified by a software to digitalize the values & compare them statistically, hence obtain solid idea about the significance of the study

2- Fig 7: he * is comparison to what group?? also figure 8??

3- Fig 10: stainig should be analyzed by an image analysis software and quantified then statistically compared.

4- Figure 13:  is not complete & control group should be presneted at the 3 timepoints/

5- Figure 13: images are poor and does not indicate a solid idea as it lacks any quantification or statistical analysis, some meaningless.

Author Response

Thanks for authors for revising the manuscript however, I think some revisions are still mandatory & authors did not reply to the previous questions & the same problems in figures are persist + colored columns have no clear axis or good resolution to show writing.

Thanks for your observations. In the present review of the article, we change all the figures for others of higher quality. We have also changed the graphs, placing the axes and making their titles bigger.

1- Figures 1,2 and 12, were not analyzed or quantified by a software to digitalize the values & compare them statistically, hence obtain solid idea about the significance of the study

R. We appreciate your comments. We have quantified the amount of calcium deposited in figure 2 (lines 400 to 410) using an image analyzer. In addition, we have added a graph and placed the significance of the 0.125 M, 0.25, and 0.5 scaffolds concerning the control (bovine bone). In addition, we have changed the caption.

Figure 12 (lines 820 to 830) shows Warthon's gelatin mesenchymal cell culture on decellularized scaffolds at 0.125 M, 0.25 M, and 0.5 M HCl concentrations. An image analyzer was used to quantify the number of cells on the scaffolding on different days. The corresponding graph was placed, and the standard deviations and significance were expressed against the control group (commercial scaffold) and each experimental group.

Concerning figure 1 (lines 341 to 374), we do not consider it necessary to quantify the cells since in the experimental groups (figure 1, a, b and c), with DAPI staining, there are no nuclei. Only controls show some cells. However, there is no presence of cells in any experimental group after the decellularization process. Therefore, it is not relevant to quantify.

2- Fig 7: he * is comparison to what group?? also figure 8??

R. In figure 7 (lines 638 to 645), we have modified the graph to give it clarity, and we place the significance of the swelling of the scaffolds concerning the control and between the different groups. We put the significance in the figure caption and modified it.

In graph 8 (lines 668 to 674), we change it to graph weight percentage versus days in the degradation buffer. Furthermore, significance was placed against the control group (commercial scaffold) and between the experimental groups. In addition, the caption was changed.

3- Fig 10: staining should be analyzed by an image analysis software and quantified then statistically compared.

R. Figure 10 (lines 753 to 761) shows the immunoexpression of the CD51 antigen, which identifies the ligand of the RGD motif in collagen. Image analysis was performed to quantify by pixels the presence of the mark observed with the antibody. The results were plotted, and the significance was added against the control and between the different experimental groups.

4- Figure 13: is not complete & control group should be presented at the 3 timepoints/

5- Figure 13: images are poor and does not indicate a solid idea as it lacks any quantification or statistical analysis, some meaningless.

R. Answer to questions 4 and 5:

Figure 13 examined the ability of scaffolds obtained with different concentrations of HCl to induce calcium deposits after being seeded with mesenchymal cells in the absence of a mineralization-inducing medium. These results are significant because they show that the 0.125 M and 0.25 M scaffolds at 28 days of culture can differentiate mesenchymal stem cells into osteoblasts in the absence of a mineralizing inducing medium.

The scaffolds cultured with stem cells were analyzed at 14 and 28 days of culture in the experiment. We welcome referee 1's suggestion, and an image of stem cell culture on decalcified bone scaffolds at 28 days was added.

On the other hand, the image analysis of each condition was performed, and the significance was plotted against the controls and between the experimental groups.

Reviewer 2 Report

I appreciate the authors' efforts to improve the manuscript, sufficiently to warrant publication in Polymers.

Author Response

We sincerely appreciate your comments and invaluable help in improving the manuscript.

Round 3

Reviewer 1 Report

I recommend acceptance of the current form of the paper.